# WILDACTOR: Unconstrained Identity-Preserving Video Generation

**Qin Guo** [1] [*]   **Tianyu Yang** [2]   **Xuanhua He** [1]   **Fei Shen** [3]   **Yong Zhang** [2] [†]   **Zhuoliang Kang** [2]
**Xiaoming Wei** [2]   **Dan Xu** [1] [‡]

**Project Page:** https://wildactor.github.io/

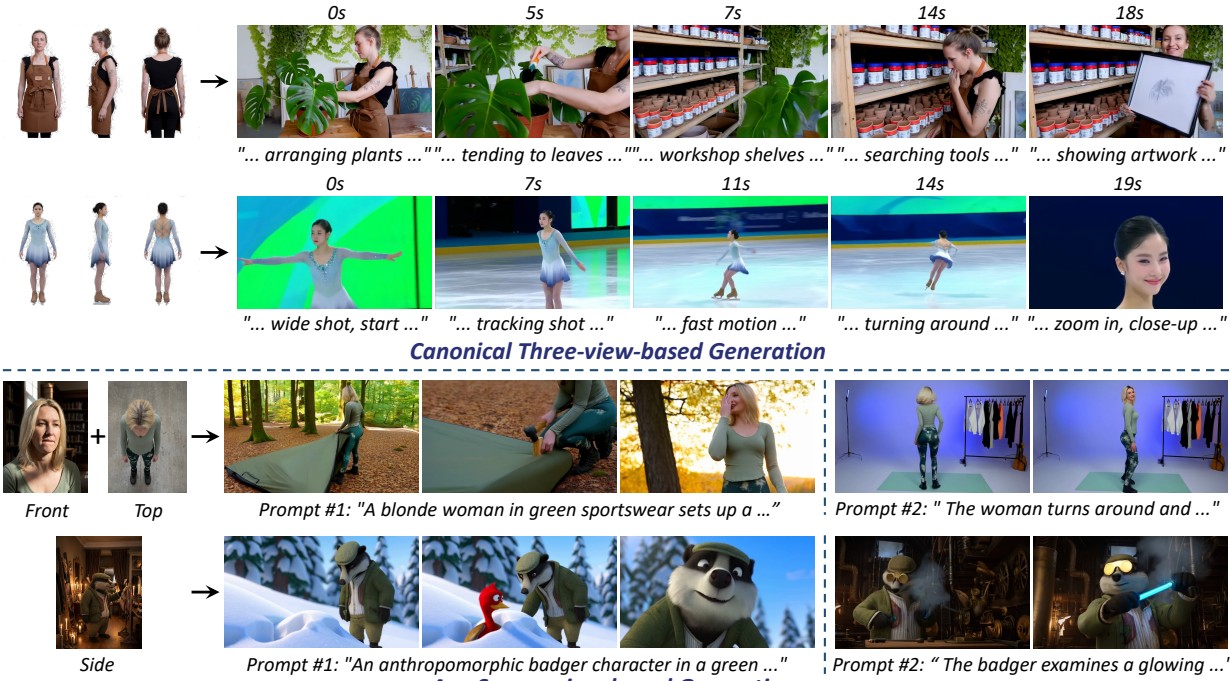

*Figure 1.* **Top:** Long-form human-centric narrative generated by WILDACTOR conditioned on canonical three-view reference images. The model preserves consistent full-body identity across viewpoint changes, camera distance variations, and dramatic motions. **Bottom:** Given any sparse-view reference images, WILDACTOR generates videos of the same subject under diverse environments, viewpoints, and motions while maintaining full-body identity consistency.

## Abstract

Production-ready human video generation requires digital actors to maintain strictly consistent full-body identities across dynamic shots, viewpoints and motions, a setting that remains challenging for existing methods. Prior methods often suffer from *face-centric* behavior that neglects body-level consistency, or produce *copy-paste* artifacts where subjects appear rigid due to pose locking. We present `Actor-18M`, a large-scale human video dataset designed to capture identity consistency under unconstrained viewpoints and environments. `Actor-18M` comprises 1.6M videos with 18M corresponding human images, covering both arbitrary views and canonical three-view representations. Leveraging `Actor-18M`, we propose WILDACTOR, a framework for any-view conditioned human video generation. We introduce an *Asymmetric Identity-Preserving Attention* mechanism coupled with a *Viewpoint-Adaptive Monte Carlo Sampling* strategy that iteratively re-weights reference conditions by marginal utility for balanced manifold coverage. Evaluated on the proposed `Actor-Bench`, WILDACTOR consistently preserves body identity under diverse shot compositions, large viewpoint

[*]Work done during an internship at Meituan. [†]Project Leader. [‡]Corresponding Author. [1]The Hong Kong University of Science and Technology [2]Meituan [3]National University of Singapore. Correspondence to: Dan Xu <danxu@cse.ust.hk>.

*Proceedings of the 43rd International Conference on Machine Learning*, Seoul, South Korea. PMLR 306, 2026. Copyright 2026 by the author(s).

transitions, and substantial motions, surpassing existing methods in these challenging settings.

## 1. Introduction

In professional cinematography, the physical permanence of an actor serves as the anchor of visual storytelling. This fundamental principle requires maintaining a strictly invariant identity across shots, viewpoints, and motions. However, replicating this **invariant identity** in video generation community remains an open challenge. While recent Diffusion Transformers (DiT) models (OpenAI, 2024; Kuaishou Team, 2024; Bao et al., 2024; Wan et al., 2025; Jiang et al., 2025; Xue et al., 2025) achieve remarkable photorealism, they often struggle to maintain identity consistency across changing viewpoints. As viewpoints shift, we frequently observe *identity drift*, where facial features degrade or clothing textures change in ways inconsistent with the subject.

Current approaches suffer from two primary limitations. First, they are predominantly *face-centric* or rely on naive injection. Methods adapting face recognition encoders (He et al., 2024; Wei et al., 2025; Yuan et al., 2025b) tend to overemphasize discriminative facial cues (e.g., hairline) while ignoring the body. This results in a "floating head" effect where the body creates hallucinations. Conversely, methods that encode the entire reference image via naive concatenation (Jiang et al., 2025) often induce *pose locking*. The generator treats the reference pose as the canonical view, restricting the synthesized character's motion and leading to static, "copy-paste" artifacts. Second, there is a critical lack of large-scale datasets for learning view-invariant human representations in the wild. Prior attempts like *Virtually Being* (Xu et al., 2025) rely on expensive studio captures, which limits their scalability to real world environments.

To address these challenges, we introduce Actor-18M, a large-scale human video dataset comprising over 1.6M high-quality videos and 18M corresponding human images. Unlike generic collections, Actor-18M provides multiple reference images of the same subject spanning diverse viewpoints, environments, and motions within videos, enabling models to learn identity-consistent representations under unconstrained conditions.

Leveraging Actor-18M, we propose WILDACTOR, a framework for any-view conditioned human video generation. It introduces an Asymmetric Identity-Preserving Attention mechanism, which enforces a unidirectional flow where video tokens query identity cues while reference tokens remain isolated from noisy backbone features, thereby preserving identity fidelity without interfering with the backbone representation. To further enhance identity injection, we propose a Viewpoint-Adaptive Monte Carlo Sampling strategy that re-weights reference images by marginal util-

ity, encouraging complementary viewpoint coverage during training. Together, these designs enable robust identity preservation under large viewpoint variations. Consequently, WILDACTOR outperforms existing methods on Actor-Bench across identity preservation and semantic alignment, while results in Fig. 1 demonstrate stable human-centric narrative and multi-shot generation under substantial changes in environment, viewpoint, and action.

In summary, our contributions are:

- We curate Actor-18M, a large-scale human video dataset comprising 1.6M videos and 18M corresponding human images, providing diverse identity references across viewpoints, environments, and motions.

- We propose WILDACTOR, a unified framework with an Asymmetric Identity-Preserving Attention mechanism and a Viewpoint-Adaptive Monte Carlo Sampling strategy, enabling robust any-view conditioning without compromising backbone representations.

- We establish Actor-Bench, on which WILDACTOR consistently outperforms prior methods in narrative coherence and contextual generalization under large viewpoint and motion variations.

## 2. Related Work

**Identity-Preserving Video Generation.** The field of video generation has rapidly transitioned from U-Net architectures to DiT (Peebles & Xie, 2023). Foundation models such as Sora (OpenAI, 2024), Kling (Kuaishou Team, 2024), and Wan (Wan et al., 2025) have demonstrated exceptional temporal coherence and photorealism. However, maintaining strict subject identity control in these models remains a significant challenge. Recent research has focused on end-to-end Subject-to-Video (S2V) frameworks to address this. Methods like VACE (Jiang et al., 2025) and Phantom (Liu et al., 2025) attempt to align reference subjects with text prompts through cross-modal mechanisms. Stand-In (Xue et al., 2025) and BindWeave (Li et al., 2025) further advance this direction by introducing dedicated modules to improve identity preserving. Despite these improvements, existing subject-driven video models still suffer from two common failure modes. First, they often exhibit *copy-paste* artifacts, where the injected subject appears static and fails to follow motion prompts. Second, they are prone to *identity drift* under viewpoint changes, where facial features or body appearance become inconsistent due to rigid identity encoding and limitations in training data. In a different direction, Virtually Being (Xu et al., 2025) utilizes multi-view studio captures to ensure view-invariant consistency. While effective, its reliance on expensive volumetric data limits its applicability to in-the-wild synthesis. In contrast,

WILDACTOR is designed as a robust framework that leverages large-scale in-the-wild data to achieve view-invariant human generation without complying with the limitations of expensive studio data.

**Human-Centric Video Datasets.** Existing human-related datasets (Zhu et al., 2022; Shen et al., 2025; Wang et al., 2024b; Guo et al., 2025; Yuan et al., 2025a) exhibit notable limitations for robust human-centric video generation. Talking-head datasets such as CelebV-HQ (Zhu et al., 2022) and TalkingFace-Wild (Shen et al., 2025) focus on facial dynamics but lack body motion and appearance. Human-centric datasets like HumanVid (Wang et al., 2024b) primarily support pose-driven generation, while providing limited identity-related annotations. More recently, OpenS2V-Nexus (Yuan et al., 2025a) introduces a large-scale dataset for subject-driven generation across diverse categories, but it is not tailored to humans and lacks fine-grained annotations essential for human video synthesis, such as camera viewpoints, lighting conditions, and canonical views. To address these limitations, we introduce `Actor-18M`, a human-centric dataset that provides identity-consistent annotations across diverse viewpoints, environments, and lighting conditions, enabling robust learning of view-invariant human representations.

# 3. The Proposed `Actor-18M` Dataset

As discussed in Sec. 2, existing datasets fail to provide consistent multi-view information for humans in videos, limiting generation capabilities in complex scenarios. To address this, we construct `Actor-18M`, a large-scale dataset containing 1.6M identity-consistent videos annotated with over 18M reference images. It provides dense supervision across diverse environments, viewpoints, and motions. **Detailed construction pipelines, model specifications, and hyperparameter settings are provided in Apx. B.**

*Table 1.* **Detailed statistics of `Actor-18M`.** *Self-Crop*: references cropped directly from raw videos. *View-Aug* and *Attr-Aug*: generated data for viewpoint and attribute augmentation, respectively. Viewpoints include Front (F), Left (L), Right (R), Up (U), Down (D) for faces, and Front (F), Side (S), Back (B) for bodies. The generated data substantially mitigates the strong frontal-view bias.

| Subset | Region | Source | Quantity | Viewpoint Distribution (%) |
|---|---|---|---|---|
| A | Face | Self-Crop | 1.05M | F:30.8 / L:16.3 / R:25.0 / U:2.9 / D:25.0 |
| | | View-Aug | 5.72M | F:8.6 / L:30.2 / R:40.9 / U:10.0 / D:10.3 |
| | Body | Self-Crop | 1.64M | F:62.8 / S:36.6 / B:0.6 |
| | | View-Aug | 8.73M | F:26.2 / S:71.6 / B:2.2 |
| B | Face | Self-Crop | 149K | F:33.9 / L:18.5 / R:25.1 / U:8.2 / D:14.3 |
| | | Attr-Aug | 281K | F:68.4 / L:2.8 / R:21.2 / U:3.9 / D:3.7 |
| | Body | Self-Crop | 149K | F:65.9 / S:34.0 / B:0.1 |
| | | Attr-Aug | 250K | F:65.9 / S:31.5 / B:2.6 |
| C | 3-View | – | 10K | – |
| Total | Face | Self-Crop | 1.20M | F:31.7 / L:16.7 / R:25.0 / U:3.3 / D:23.3 |
| | | Generated | 6.01M | F:11.4 / L:29.0 / R:40.0 / U:9.6 / D:10.0 |
| | Body | Self-Crop | 1.79M | F:63.1 / S:36.3 / B:0.5 |
| | | Generated | 8.98M | F:27.3 / S:70.5 / B:2.2 |

## 3.1. Data Collection and Filtering

We collect videos from internal high-quality sources and OpenS2V (Yuan et al., 2025a). To ensure identity consistency, we implement a two-stage filtering pipeline. First, we apply a coarse filter by sampling sparse frames and computing facial similarity (Deng et al., 2019) to remove obvious identity shifts. Second, we perform fine-grained filtering using dense point tracking (Karaev et al., 2024) and clip similarity verification (Radford et al., 2021) to ensure strict subject consistency across frames. This process yields 1.6M single-person videos.

## 3.2. Data Construction

We detect and segment face and body regions using segmentation models (Deng et al., 2020; Yu et al., 2018; Cheng et al., 2024; Ravi et al., 2025). To decouple identity from environments and viewpoints, we augment the dataset into three subsets (`Actor-18M -A, -B, -C`). Fig. 2 illustrates the construction pipeline of each subset.

**`Actor-18M -A`.** To address the "pose-locking" issue caused by correlated viewpoints, we construct `Actor-18M -A` by generating view-transformed references. Using a multi-angle image editing model (HuggingFace User dx8152, 2025), we synthesize face and body images from six diverse viewing angles for each subject. We verify the results using a Multimodal LLM (MLLM) (Bai et al., 2025) to ensure that identity and clothing remain consistent despite substantial viewpoint changes.

**`Actor-18M -B`.** To prevent overfitting to background or lighting, `Actor-18M -B` introduces attribute diversification. We define an attribute pool covering 200 environments, 8 expressions, 10 lighting conditions, and 30 motions. We employ an MLLM (Bai et al., 2025) to generate editing instructions based on these attributes, followed by an image editing model (Wu et al., 2025) to synthesize new references. This process yields images with diverse styles while preserving the subject's identity.

**`Actor-18M -C`.** To provide complete identity anchors, `Actor-18M -C` contains canonical three-view images (front, side, back). We filter videos to identify subjects observable from all three viewpoints using pose estimation (Yang et al., 2023) and MLLM refinement (Bai et al., 2025). Using high-visibility frames as prompts, we generate canonical character sheets using a reference-based generation model (Comanici et al., 2025).

Finally, we annotate all references with fine-grained metadata, including orientation and visibility ratios.

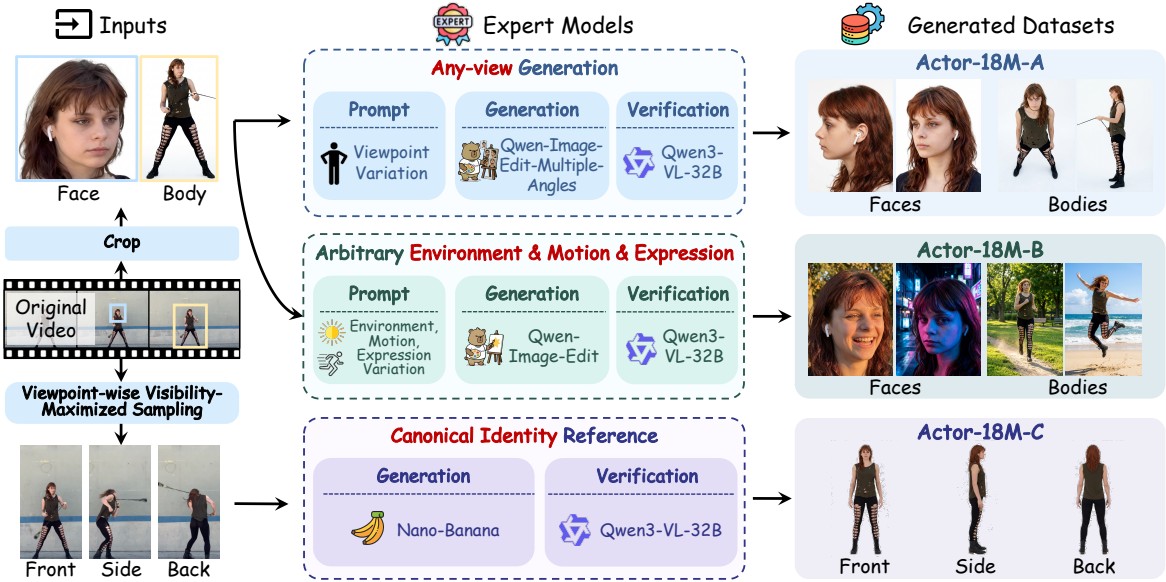

*Figure 2.* **Construction pipeline and representative samples of `Actor-18M`.** *Left:* Frames are sampled and filtered from identity-consistent videos, from which facial and body images are extracted as ground-truth references. *Right:* Based on these references, view-transformed samples are generated to construct `Actor-18M-A`, while attribute-conditioned image editing under diverse environments, lighting conditions, and motions produces `Actor-18M-B`. Canonical three-view images in `Actor-18M-C` are generated using Nano-Banana, guided by frames with the highest visibility selected from different viewpoints, serving as complete identity anchors.

### 3.3. Data Statistics

Tab. 1 presents the detailed statistics. In total, `Actor-18M` comprises 1.6M videos accompanied by over 18M reference images, providing dense supervision for identity preservation. A critical issue in raw data is severe viewpoint imbalance, evidenced by a strong frontal bias (e.g., 63.1% frontal bodies). To address this, `Actor-18M-A` explicitly augments viewpoint diversity, shifting the body distribution substantially toward side views (70.5%) to yield more uniform coverage across all viewing angles. Similarly, facial data is enriched with non-frontal views, where profile views account for over 69% of generated samples, compared to only 42% in original sources. Complementary to this, `Actor-18M-B` and `-C` further expand attribute diversity and provide canonical anchors, establishing a robust foundation for unconstrained generation.

## 4. The Proposed Method

We address the task of any-view-conditioned human video generation. Given a textual prompt $\mathcal{C}_{\text{txt}}$, a set of $N$ reference facial images $\mathcal{I}^f = \{\mathcal{I}^f_k\}_{k=1}^N$, and a set of $M$ reference body images $\mathcal{I}^b = \{\mathcal{I}^b_k\}_{k=1}^M$ captured under diverse viewpoints, environments, and poses, the goal is to generate a video $\mathcal{V}$ that aligns with the textual prompt while faithfully preserving human identity with high fidelity across viewpoints and maintaining temporal coherence.

### 4.1. Preliminaries

Our method builds upon a latent video DiT (Meituan Long-Cat Team et al., 2025) trained with Rectified Flow (RF) (Lipman et al., 2023). Given a video clip $\mathbf{V} \in \mathbb{R}^{T \times H \times W \times 3}$, we encode it into a latent representation $\mathbf{z}_0 := \mathcal{E}(\mathbf{V}) \in \mathbb{R}^{f \times c \times h \times w}$ using the encoder $\mathcal{E}$ of a variational autoencoder (VAE) (Kingma & Welling, 2014), with the decoder denoted as $\mathcal{D}$. RF defines a linear interpolation between the data $\mathbf{z}_0 \sim p_{\text{data}}$ and a Gaussian prior sample $\boldsymbol{\epsilon} \sim \mathcal{N}(\mathbf{0}, \mathbf{I})$, defined as $\mathbf{z}_t := (1-t)\mathbf{z}_0 + t\boldsymbol{\epsilon}$ with $t \in [0, 1]$, which induces a constant velocity field $\boldsymbol{\epsilon} - \mathbf{z}_0$. A video DiT $\mathbf{v}_{\boldsymbol{\theta}}$ is trained to predict this velocity using the objective:

$$\mathcal{L}_{\text{RF}} := \mathbb{E}_{t, \mathbf{z}_0, \boldsymbol{\epsilon}} \left[ w(t) \left\| \mathbf{v}_{\boldsymbol{\theta}}(\mathbf{z}_t, t, \mathbf{C}_{\text{ctx}}) - (\boldsymbol{\epsilon} - \mathbf{z}_0) \right\|_2^2 \right], \tag{1}$$

where $w(t)$ is a time-dependent weighting function. In our setting, the condition $\mathbf{C}_{\text{ctx}}$ is defined as $\{\mathcal{C}_{\text{txt}}, \mathcal{I}^f, \mathcal{I}^b\}$, which aggregates the prompt together with reference facial and body images for conditional video generation.

### 4.2. Model Design

WILDACTOR presents a framework that injects multi-view facial and body cues into a video DiT backbone. To decouple identity information from backbone representations, we introduce an *Asymmetric Identity-Preserving Attention (AIPA)* mechanism. In addition, we propose an *Identity-Aware 3D RoPE (I-RoPE)* scheme to explicitly distinguish video tokens from identity reference tokens.

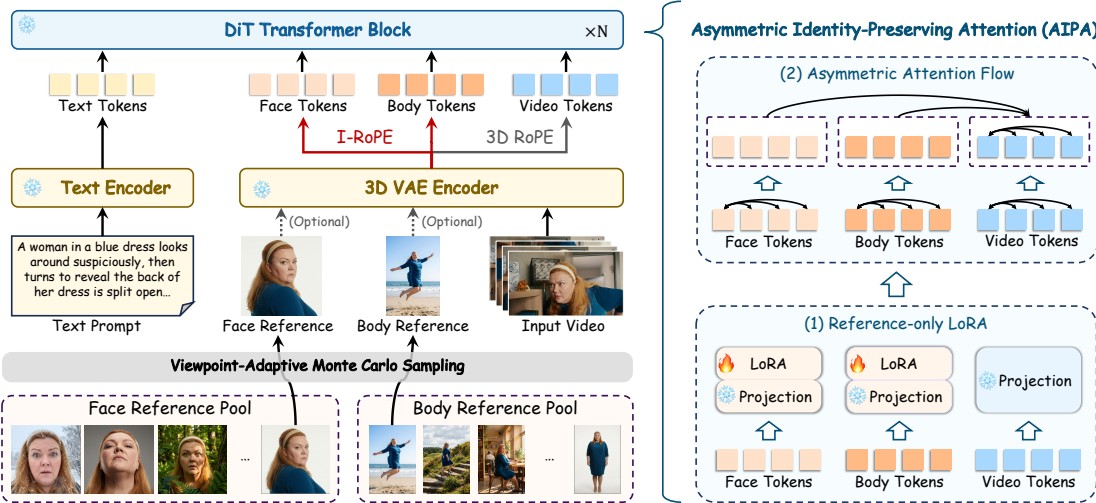

*Figure 3.* **Overview of WILDACTOR.** *Left:* The overall architecture, where a video DiT is conditioned on multi-view facial and body reference images selected via Viewpoint-Adaptive Monte Carlo Sampling. Reference tokens are embedded with I-RoPE to distinguish them from video tokens in the shared spatio-temporal attention space. *Right:* Details of AIPA, illustrating how identity reference tokens are aggregated and injected into video tokens through asymmetric attention while preserving prior of backbone.

**Asymmetric Identity-Preserving Attention.** Naïvely applying full attention across video and reference tokens often leads to identity leakage, where static appearance cues dominate motion generation and result in pose-locking artifacts. AIPA addresses this issue by enforcing an asymmetric information flow, in which reference tokens provide identity cues to video tokens while remaining isolated from noisy video latents. This mechanism consists of two key components: *(1) reference-only LoRA* and *(2) asymmetric attention flow*.

*(1) Reference-only LoRA.* Instead of adapting the full backbone, we apply lightweight LoRA modules exclusively to reference tokens. Let $\mathbf{W}_{Q,K,V}$ denote the frozen projection matrices of the self-attention in DiT. For reference tokens $\mathbf{c} \in \{\mathbf{f}_{\text{face}}, \mathbf{f}_{\text{body}}\}$, the projections are computed as:

$$\mathbf{q}_c, \mathbf{k}_c, \mathbf{v}_c = (\mathbf{W}_{Q,K,V} + \Delta\mathbf{W}^{\text{ref}}_{Q,K,V})\mathbf{c}, \qquad (2)$$

where $\Delta\mathbf{W}^{\text{ref}}_{Q,K,V}$ are learnable LoRA parameters shared by both facial and body references. In contrast, video tokens $\mathbf{z}_t$ always use the frozen backbone weights.

*(2) Asymmetric attention flow.* Within each transformer block, attention is computed in two stages. First, reference tokens perform self-attention independently to aggregate multi-view facial and body information into a unified identity representation $\mathbf{C}_{\text{ref}}$, while video tokens simultaneously undergo standard self-attention to model temporal dynamics. Second, an asymmetric fusion is applied, where video tokens act as Queries and attend to both video and reference tokens, with Keys and Values formed as:

$$Q = \mathbf{z}_t, \quad K = [\mathbf{z}_t; \mathbf{C}_{\text{ref}}], \quad V = [\mathbf{z}_t; \mathbf{C}_{\text{ref}}].$$

**I-RoPE.** Since video tokens and identity reference tokens share the same attention context, assigning identical positional encodings can cause ambiguity between temporal motion and static appearance. I-RoPE resolves this by assigning distinct spatio-temporal coordinates to different token types: *(1) Temporal separation.* Video tokens follow the canonical temporal indices $t \in [0, T]$, while reference facial and body tokens are assigned fixed temporal offsets, $T + \Delta_f$ and $T + \Delta_b$, respectively, where $\Delta_f = 4$ and $\Delta_b = 128$. *(2) Spatial separation.* In standard 3D RoPE, video tokens are encoded with positional tuples $(t, h, w)$, where $h \in [0, H]$ and $w \in [0, W]$. To spatially decouple identity references from video tokens, we shift the spatial coordinates of reference facial and body tokens so that their $(h, w)$ indices start from $(H_{\max}, W_{\max})$, where $H_{\max}$ and $W_{\max}$ denote the maximum spatial dimensions of the video sequence. This ensures that reference tokens occupy distinct spatial positions in the joint spatio-temporal embedding space while preserving the original 3D RoPE formulation.

### 4.3. Training Scheme

To effectively leverage the multi-view information of Actor-18M, we design a training scheme that promotes complementary reference sampling, encouraging diverse and informative viewpoints to be jointly observed during training for robust any-view conditioning.

**Viewpoint-Adaptive Monte Carlo Sampling.** Naïve sampling on Actor-18M -A and -B often yields redundant views (e.g., multiple frontal images), hindering the effective utilization of multi-view information. To address this, we employ a dynamic re-weighting strategy to encourage diverse coverage. Specifically, given a candidate set $\mathcal{S}$, each

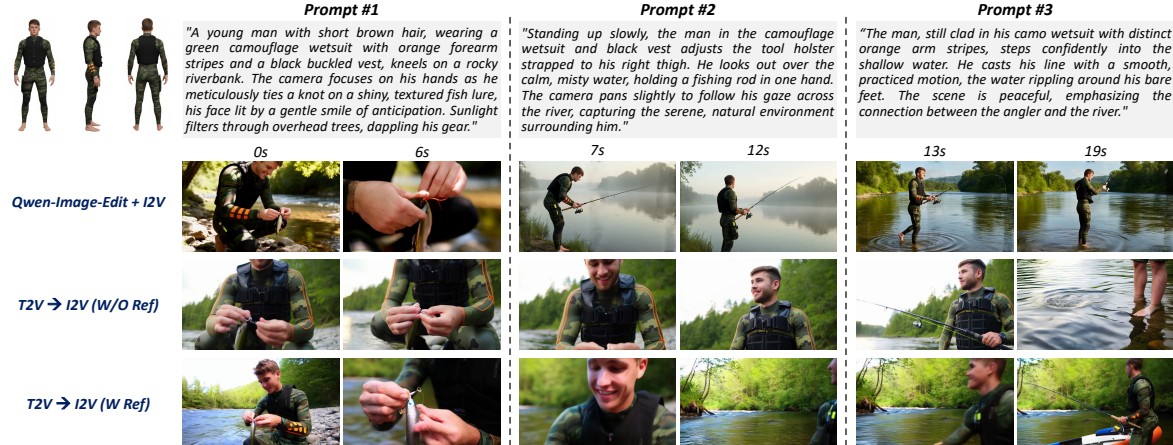

Figure 4. **Qualitative comparison on sequential narrative.** WILDACTOR maintains stronger full-body consistency and prompt adherence than prior methods under viewpoint changes, camera motion, and scene transitions. **Zoom in to better compare fine-grained details.**

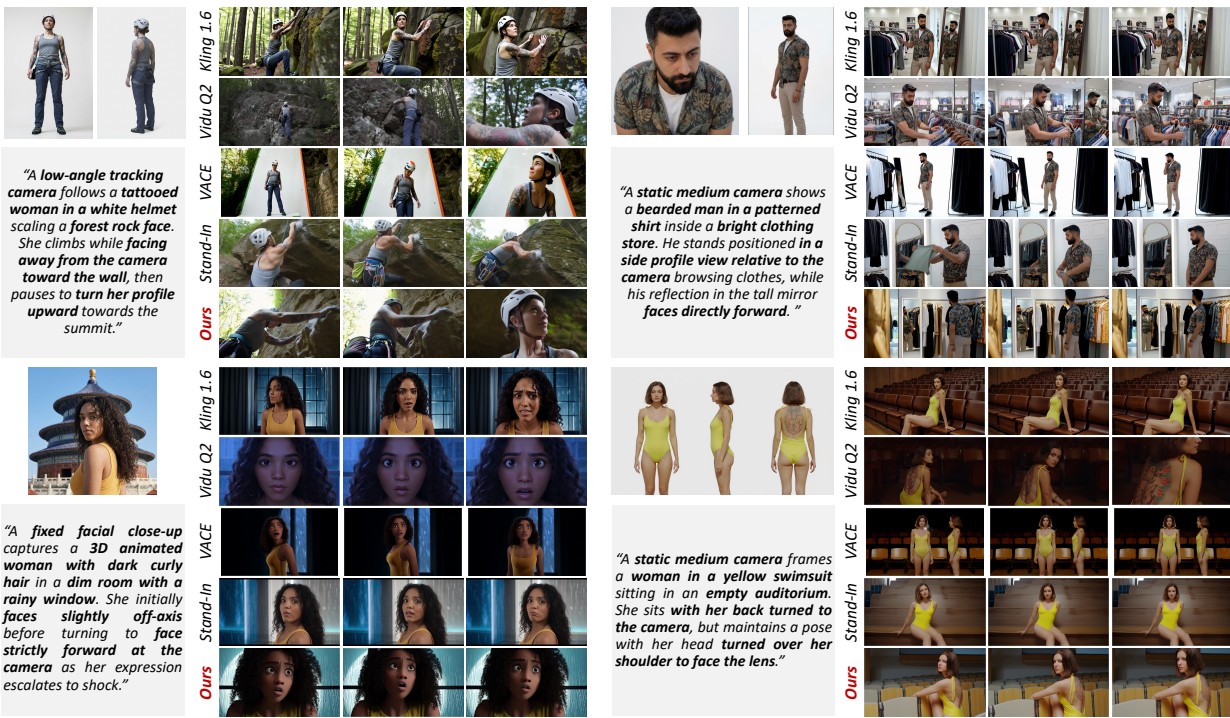

Figure 5. **Qualitative comparisons with representative open-source and commercial models on *contextual generalization*.** Our approach demonstrates superior full-body consistency and prompt adherence, particularly under challenging viewpoint and camera changes. Fine-grained appearance details are better preserved. **Zoom in to better compare fine-grained details.**

image is assigned an initial weight. Upon sampling a reference $x^*$, we suppress remaining candidates located within its angular neighborhood $|\theta_{x^*} - \theta_{x_j}| < \delta$ by updating their weights according to the rule:

$$w_j \leftarrow w_j \cdot \gamma, \tag{3}$$

where $\gamma < 1$ is a decay factor. This adaptive suppression biases the process toward complementary viewpoints, enabling the model to observe a more uniform distribution of the identity manifold.

## 5. Experiments

### 5.1. Implementation Details

Our framework is built upon an internal 5B DiT model with an architecture similar to LongCat-Video (Meituan LongCat Team et al., 2025). We fine-tune the model using rank-128 LoRA applied to the QKV and output projection layers of self-attention, introducing 0.29B trainable parameters. Training is performed in a mixed T2V, I2V, and VC tasks using the AdamW optimizer with a learning rate of $1 \times 10^{-4}$

on 16 NVIDIA H100 GPUs. We train the model for 80K steps with two stages: $256p$ for the first 70K steps and $480p$ for the remaining 10K steps. All videos contain 93 frames. During training, 0–5 reference facial and body images are randomly sampled, with data drawn from `Actor-18M -A`, `-B`, and `-C` in a 5:2:1 ratio. For the Viewpoint-Adaptive Sampling strategy, we set the angular neighborhood threshold $\delta$ to $\pi/6$ and the decay factor $\gamma$ to 0.5, ensuring a balanced trade-off between diversity and sampling efficiency.

## 5.2. The Proposed `Actor-Bench`

To comprehensively evaluate our model , we establish `Actor-Bench`, which consists of 75 distinct subjects evenly divided into three conditioning settings: *canonical three-view*, *arbitrary viewpoint*, and *in-the-wild*. For each subject, manually verified canonical three-view references are provided to ensure reliable evaluation.

**Evaluation Axes.** For each subject, we evaluate the model along two complementary axes. *(1) Sequential narrative.* Three consecutive prompts planned by Gemini-3-Pro (Comanici et al., 2025) that form a coherent storyline, used to evaluate identity consistency in long-form videos, while covering diverse actions and viewpoints. *(2) Contextual generalization.* Prompt generated by Gemini-3-Pro that describes subjects under diverse environments, viewpoints, and motions, designed to assess generalization under real-world scenarios.

**Evaluation Metrics.** We evaluate generated videos from three complementary aspects. Specific calculation protocols for the VLM-based metrics are detailed in Apx. C. *(1) Body consistency.* Following VBench2 (Zheng et al., 2025), we assess body-level identity consistency using Gemini-3-Pro. To ensure robustness against viewpoint changes, we employ a viewpoint-aware verification pipeline that matches generated frames with ground-truth references of the closest viewpoint. *(2) Face identity preservation.* We measure facial identity consistency by computing the cosine similarity of ArcFace embeddings on detected faces. Since ArcFace similarity is sensitive to viewpoint changes, each generated face is matched with the ground-truth reference from the most similar viewpoint to ensure fair comparison. *(3) Semantic alignment.* We evaluate feature-level text-video semantic consistency using ViCLIP (Wang et al., 2024a). In addition, Gemini-3-Pro is employed to assess VLM-level semantic adherence, specifically checking whether the generated video holistically follows prompt-specified attributes such as appearance and viewpoint.

## 5.3. Evaluation

**Baseline Setup.** *(1) Sequential narrative.* We compare three representative strategies for long-form video generation: (a) *Qwen-Image-Edit + I2V*: each clip is initialized by Qwen-Image-Edit conditioned on the references, followed by I2V generation; (b) *Autoregressive T2V → I2V (w/o Ref)*: the first clip is generated by T2V, and subsequent clips are produced by iteratively applying I2V from the last frame; (c) *Autoregressive T2V → I2V (w/ Ref, Ours)*: identical to (b), but with identity references condition. *(2) Contextual generalization.* We compare WILDACTOR with a set of representative baselines, including open-source models VACE (Jiang et al., 2025) and Stand-In (Xue et al., 2025), as well as closed-source commercial models including Vidu Q2 (Bao et al., 2024) and Kling 1.6 (Kuaishou Team, 2024).

**Qualitative Results.** Fig. 4 and Fig. 5 present qualitative comparisons between WILDACTOR and baselines .

*Sequential narrative.* Qwen-Image-Edit + I2V often suffers from noticeable discontinuities between clips, as identity and environment are re-initialized at each segment. Naïve T2V → I2V tend to accumulate errors over time, leading to identity drift and temporal inconsistency. By incorporating identity references throughout the autoregressive generation process, WILDACTOR maintains consistent identity and coherent temporal progression over long narratives.

*Contextual generalization.* Open-source baselines such as VACE and Stand-In struggle to balance identity preservation, motion flexibility, and prompt adherence. VACE often fails to disentangle the subject from the reference layout, resulting in *copy-paste* artifacts and weak prompt following, while Stand-In exhibits severe *pose locking*, where the generated subject remains rigidly constrained to the reference viewpoint. Commercial models such as Kling 1.6 and Vidu Q2 produce visually smooth videos with strong motion dynamics, but frequently show weaker control over viewpoint transitions or prompt-specified attributes. Overall, WILDACTOR maintains more robust full-body consistency while faithfully following changes in viewpoint, camera motion, and action across diverse scenarios.

**Quantitative Results.** Tab. 2 summarizes quantitative comparisons with representative baselines.

*Sequential narrative.* The *Autoregressive T2V → I2V (w/o Ref)* baseline exhibits identity drift, achieving a Face Identity score of only 0.320. *Qwen-Image-Edit + I2V* performs better but remains substantially inferior to WILDACTOR in body consistency. WILDACTOR achieves the strongest identity fidelity and semantic alignment, effectively mitigating error accumulation in long-form generation.

*Contextual generalization.* Commercial model Vidu Q2 obtain higher face identity scores, likely due to aggressive appearance copying that favors facial similarity over structural flexibility. In contrast, WILDACTOR attains the highest body consistency score (0.952 vs. 0.905 for Vidu Q2), substantially reducing artifacts under viewpoint variation. Moreover, WILDACTOR achieves the highest VLM-level

*Table 2.* **Quantitative comparisons on `Actor-Bench`** under (1) *Sequential Narrative* and (2) *Contextual Generalization*. Best and runner-up are **bold** and underlined. * denotes closed-source commercial models.

| Method | Params | Face Identity ↑ | Body Consistency ↑ | Semantic Alignment | |
|---|---|---|---|---|---|
| | | | | Feature-Level ↑ | VLM-Level ↑ |
| *Setting 1: Sequential Narrative* | | | | | |
| Qwen-Image-Edit + I2V | 5B | 0.521 | 0.720 | 0.228 | 0.733 |
| T2V → I2V (*w/o* Ref) | 5B | 0.320 | 0.450 | 0.215 | 0.613 |
| T2V → I2V (*w* Ref, Ours) | 5B | **0.548** | **0.925** | **0.235** | **0.893** |
| *Setting 2: Contextual Generalization* | | | | | |
| VACE (Jiang et al., 2025) | 14B | 0.485 | 0.582 | 0.221 | 0.667 |
| Stand-In (Xue et al., 2025) | 14B | 0.510 | 0.416 | 0.218 | 0.600 |
| Vidu Q2* (Bao et al., 2024) | – | **0.565** | 0.905 | **0.241** | 0.880 |
| Kling 1.6* (Kuaishou Team, 2024) | – | 0.558 | 0.885 | 0.239 | 0.867 |
| WILDACTOR | 5B | 0.559 | **0.952** | 0.238 | **0.920** |

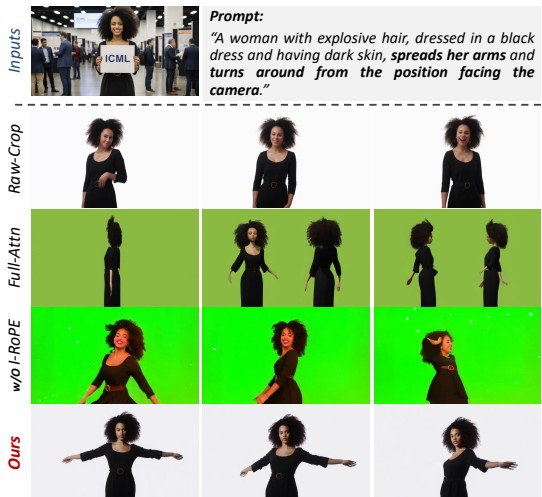

*Figure 6.* **Ablation study on data strategies and model components.** We evaluate variants in controlled scenes with turning motions, enabling clear comparison under viewpoint changes.

*Table 3.* **Ablation of dataset strategies.** We report Body Consistency scores across different target viewpoints.

| Setting | Body Consistency ↑ | | | |
|---|---|---|---|---|
| | Front | Side | Back | Average |
| Raw-Crop | 0.885 | 0.725 | 0.680 | 0.802 |
| Random Sampling | 0.915 | 0.840 | 0.785 | 0.865 |
| Viewpoint-Adaptive | **0.958** | **0.952** | **0.937** | **0.952** |

*Table 4.* **Ablation of model components.** We analyze the impact of AIPA and I-RoPE on identity and semantic alignment.

| Setting | Component | | Face ID ↑ | Body Cons. ↑ | Semantic Alignment | |
|---|---|---|---|---|---|---|
| | AIPA | I-RoPE | | | Feature-Level ↑ | VLM-Level ↑ |
| Full-Attn | ✗ | ✓ | 0.515 | 0.890 | 0.215 | 0.610 |
| *w/* AIPA only | ✓ | ✗ | 0.542 | 0.825 | 0.230 | 0.895 |
| WILDACTOR | ✓ | ✓ | **0.559** | **0.952** | **0.238** | **0.920** |

score, indicating better faithfulness to complex prompts.

### 5.4. Ablation Study

We conduct ablation studies on *data strategies* and *model designs*. Tab. 3, Tab. 4 and Fig. 6 demonstrate the results.

**Effect of Dataset and Sampling Strategy.** We compare three settings to validate our data construction and sampling: *(1) Raw-Crop*: training using identity references cropped directly from raw training videos; *(2) Random Sampling*: training on `Actor-18M` with random sampling; *(3) Viewpoint-Adaptive*: training on `Actor-18M` with our adaptive sampling. As presented in Tab. 3, the *Raw-Crop* baseline performs robustly on frontal views. However, it suffers from noticeable degradation on side and back views due to the severe viewpoint imbalance in raw data. Incorporating `Actor-18M` with *Random Sampling* mitigates this issue, boosting the average consistency. Furthermore, our *Viewpoint-Adaptive* strategy enables the model to learn viewpoint-agnostic robustness. This elevates the average score to 0.952, with high consistency maintained even on the most challenging back view scenarios.

**Effect of AIPA.** We replace AIPA with standard full self-attention (Full-Attn) while retaining I-RoPE. As shown in Tab. 4 (Row 1 vs. Row 3), the Full-Attn baseline suffers from a significant drop in semantic adherence. This indicates that standard attention causes the injected reference features to conflict with the textual control. In contrast, our AIPA mechanism improves facial identity preservation while maintaining strong instruction adherence.

**Effect of I-RoPE.** We remove I-RoPE to assess the impact of positional differentiation. As shown in Tab. 4 (Row 2 vs. Row 3), this leads to confusion between reference and video features, resulting in a sharp decline in body consistency. By explicitly encoding the distinction, I-RoPE ensures robust structural coherence and motion quality.

## 6. Conclusion

In this paper, we introduce `Actor-18M`, a large-scale human-centric video dataset that provides diverse and complementary identity references across unconstrained environments, viewpoints, and motions, addressing a key data limi-

tation in identity-consistent human video generation. Building on `Actor-18M`, we present WILDACTOR, a framework for any-view conditioned human video generation that consistently preserves full-body identity across dynamic shots, large viewpoint transitions, and substantial motions, which remain challenging for existing methods.

## Impact Statement

This paper presents work whose goal is to advance the field of identity-preserving video generation. There are many potential societal consequences of our work, and we strongly advocate for the responsible and ethical application of this technology.

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

## A. More Results

For additional qualitative results, please refer to the `index.html` provided in the supplementary material.

## B. Implementation Details of Dataset Construction

In this section, we provide a comprehensive description of the engineering pipeline used to construct `Actor-18M`. We detail the model specifications, hyperparameter settings, and algorithmic protocols for data filtering, annotation, and generative augmentation.

### B.1. Data Processing Pipeline

To process the massive scale of raw videos, we design a robust pipeline that ensures both identity consistency and annotation precision.

**Stage 1: Cascaded Filtering Strategy.** We employ a coarse-to-fine strategy to efficiently filter 1.6M high-quality videos from raw sources.

- **Coarse Filtering (Identity Stability)**: We first process videos at a sparse sampling rate of 1 fps. For each video, we extract faces and compute the cosine similarity between the first frame and subsequent frames using ArcFace (Deng et al., 2019). Videos with an average similarity score below 0.4 are discarded immediately to remove obvious identity shifts or false detections.

- **Fine-Grained Filtering (Motion & Consistency)**: Passing candidates are then up-sampled to 8 fps. We employ CoTracker (Karaev et al., 2024) to generate dense point tracks on the subject. We analyze these tracks to filter out videos with severe occlusions or rapid camera cuts. Additionally, we compute CLIP-based (Radford et al., 2021) frame-to-frame similarity. Only videos maintaining a CLIP consistency score above 0.45 are retained, ensuring the subject's appearance remains stable under motion.

**Stage 2: Automated Annotation Pipeline.** We deploy specialized models to extract precise pixel-level annotations for both face and body regions.

- **Face Region**: We utilize RetinaFace (Deng et al., 2020) for robust facial landmark detection and bounding box regression. Based on these detections, BiSeNet (Yu et al., 2018) is applied to generate fine-grained semantic segmentation masks for the face parsing task.

- **Body Region**: We adopt a two-step approach for body segmentation. First, YOLO-World (Cheng et al., 2024) is prompted with the text "person" to detect human bounding boxes. These boxes serve as prompts for the Segment Anything Model 2 (SAM2) (Ravi et al., 2025), which outputs high-quality, temporal-consistent instance masks for the entire human body.

### B.2. Generative Augmentation Pipeline

To construct the three specialized subsets (`Actor-18M -A, -B, -C`), we implement a sophisticated generative pipeline leveraging Multimodal LLMs (MLLMs) and advanced image editing models.

**Subset A: Viewpoint Transformation.** The goal of this subset is to decouple identity from camera viewpoints.

1. **Source Extraction**: We crop the clearest face and body images from the original video as source references.

2. **Multi-Angle Synthesis**: We utilize Qwen-Image-Edit-Multiple-Angles (HuggingFace User dx8152, 2025), a specialized internal model finetuned for novel view synthesis. We prompt the model to generate the source subject from six distinct viewpoints: *Front, Left-Side, Right-Side, Back, Top-Down,* and *Bottom-Up*.

3. **Verification**: To prevent identity loss during transformation (e.g., the face changing when turning sideways), we employ Qwen3-VL-32B (Bai et al., 2025) as a judge. It compares the generated view with the source reference and assigns a consistency score. Only samples with high verification confidence are included.

**Subset B: Attribute-Conditioned Editing.** This subset prevents the model from overfitting to specific backgrounds or lighting conditions.

1. **Attribute Pool Construction**: We define a structured attribute dictionary containing:

   - **200 Environments**: Ranging from diverse indoor scenes (e.g., "library", "cozy bedroom") to outdoor landscapes (e.g., "crowded market", "serene beach").
   - **8 Expressions**: Happiness, sadness, surprise, fear, anger, disgust, contempt, and neutral.
   - **10 Lighting Conditions**: Sunny day, overcast sky, golden hour, blue hour, studio softbox, hard spotlight, neon light, candlelight, firelight, and cinematic rim light.
   - **30 Motions**: Walking, running, sitting, standing, reading, drinking, eating, talking, phone calling, waving, jumping, dancing, cycling, yoga, stretching, boxing, swimming, playing basketball, playing football, hiking, singing, playing guitar, playing piano, painting, photography, laughing, crying, arguing, hugging, and shaking hands.

2. **Instruction Generation**: We randomly sample combinations from this pool and feed them into Qwen3-VL-32B. The MLLM is prompted to convert these attributes into a precise natural language editing instruction (e.g., *"Change the background to a snowy forest while keeping the person's identity unchanged..."*).

3. **Editing & Verification**: The instruction and the original image are fed into Qwen-Image-Edit (Wu et al., 2025). The resulting images undergo a rigorous CLIP-I checking loop to ensure the subject's identity embedding remains within a safe margin of the original.

**Subset C: Canonical Identity Anchors.** This pipeline reconstructs a comprehensive multi-view identity representation of the subject.

1. **Pose Filtering**: We run DWPose (Yang et al., 2023) on all raw videos to estimate head and body orientation angles. We filter for "Golden Videos" where the subject is clearly visible from all three canonical viewpoints *(Front, Side, and Back)* within a single clip.

2. **Frame Selection**: From these videos, we select three "Anchor Frames" corresponding to the peaks of the visibility scores for the three canonical views.

3. **Reference-Based Generation**: These anchor frames are used as visual prompts for Nano-Banana (Comanici et al., 2025). We instruct the model to arrange these views into a standard orthogonal character sheet.

4. **Final Audit**: Qwen3-VL-32B performs a final quality check, rejecting generations with anatomical artifacts or inconsistent clothing details.

## C. Evaluation Metrics Details

To ensure a rigorous evaluation of identity consistency and semantic alignment, we employ Gemini-3-Pro (Comanici et al., 2025) as an automated evaluator. While Body Consistency is evaluated at the frame level to capture temporal stability, Semantic Alignment is assessed at the video level to ensure holistic adherence to the text prompt.

### C.1. Body Consistency

Traditional identity metrics (e.g., ArcFace) primarily focus on facial features and are sensitive to viewpoint changes. To accurately evaluate full-body consistency under arbitrary viewpoints, we design a viewpoint-aware identity check pipeline.

Given a generated video sequence $V = \{v_t\}_{t=1}^T$ and a set of ground-truth reference images $\mathcal{I}_{ref}$ covering multiple viewpoints (Front, Side, Back), the evaluation proceeds as follows:

1. **Viewpoint Estimation**: For each frame $v_t$, Gemini-3-Pro first estimates the dominant viewpoint of the subject.

2. **Reference Matching**: The frame $v_t$ is paired with the corresponding ground-truth reference $I_{ref}^{view}$ that shares the closest viewpoint.

3. **Binary Scoring**: The VLM compares the generated subject in $v_t$ against the matched reference $I_{ref}^{view}$ to determine identity consistency. A binary score $s_t^{body} \in \{0, 1\}$ is assigned, where 1 indicates a consistent identity and 0 indicates inconsistency.

The final Body Consistency score is computed as the average of positive scores over the total number of frames:

$$\text{Score}_{\text{Body}} = \frac{1}{T} \sum_{t=1}^{T} s_t^{body} \tag{4}$$

where $T$ denotes the total number of frames in the evaluation set.

### C.2. VLM-Level Semantic Alignment

To assess how faithfully the generated videos adhere to complex textual prompts (including specific actions, environments, and lighting), we employ a video-level verification strategy.

For a generated video $V$ and its corresponding text prompt $P$, we feed the entire video clip along with the instructions into Gemini-3-Pro. The model acts as a judge to verify whether the visual content of the video holistically aligns with the attributes described in $P$.

A binary score $S_{video} \in \{0, 1\}$ is assigned for the video. The final VLM-Level Semantic Alignment score is calculated by averaging these scores across the entire test set:

$$\text{Score}_{\text{Align}} = \frac{1}{N} \sum_{i=1}^{N} S_{video}^{(i)} \tag{5}$$

where $N$ is the total number of videos in the benchmark. This video-wise assessment ensures that the model is penalized for any semantic deviations that occur throughout the generated video.

## D. Limitations

Our current implementation focuses on single-person video generation. Handling multiple subjects simultaneously requires explicitly disentangling identity features within the attention mechanism to prevent attribute mixing between different characters. Although our proposed modules (e.g., AIPA) demonstrate robust identity control, extending them to multi-person scenarios with complex interactions remains a challenging yet promising direction for future research.

