# OpenReview forum: "WildActor: Unconstrained Identity-Preserving Video Generation"
_ICML.cc/2026/Conference — ICML 2026 regular_

### Official Review · Reviewer_XJYe · 2026-02-18

**Soundness:** 3
**Presentation:** 3
**Significance:** 3
**Originality:** 2
**Overall Recommendation:** 4
**Confidence:** 5

**Summary:**

The paper introduces **Actor-18M**, a new dataset for identity-consistent long-form video generation. The dataset consists of 1.6M single-subject "in-the-wild" videos (curated from internal and public sources) and a set of accompanying reference images per video, that show the subject under unconstrained contexts, including different viewpoints, environments, clothing, poses, etc. In total, there are 18M such reference images. Using this dataset, the authors finetune a 5B DiT video model (LongCat-Video) for subject-to-video (S2V) generation, showing competitive or improved performance over recent open-source (VACE, Stand-In) and commercial (Vidu Q2, Kling 1.6) baselines. To create the synthetic reference annotations for the dataset, the authors extract selected frames from the initial 1.6M videos, and use expert models (like Nano Banana and Qwen-Image-Edit) to simulate the reference subject under different views or contexts. For the proposed DiT video model, named **WildActor**, the authors present a few slight modifications to the original backbone, like an Asymmetric Identity-Preserving Attention (AIPA) mechanism, to better inject the reference identity tokens to the generated video.

**Compliance With Llm Reviewing Policy:**

Affirmed.

**Final Justification:**

The paper introduces a new large-scale dataset for personalized video generation and presents a video DiT model that outperforms existing open-source methods in subject-to-video generation. The authors’ rebuttal adequately addresses my concerns and provides additional evidence that strengthens the paper’s contributions. The other reviewers also express positive evaluations. Therefore, I recommend acceptance.

**Key Questions For Authors:**

Below are some points that I would like to be clarified:
1.  How are the values $\Delta_f = 4$,  $\Delta_b = 128$ selected? There is no comment about it in the paper.
2. In the ablation study of Tab. 3, why is semantic alignment (Feature-Level↑,  VLM-Level↑) not evaluated? I would expect the copy-paste problem caused by non-augmented data to affect alignment with the prompt.
3. For the "Sequential Narrative" experiment, which are the T2V and I2V models used as baselines?

**Limitations:**

Yes, limitations and societal impact are mentioned.

**Strengths And Weaknesses:**

**Strengths:**
- The biggest strength of the paper is the proposed dataset. Given the scarcity of public datasets with full-body videos annotated with unconstrained references, this will be a significant contribution to the field. The quality of the dataset also looks decent.
- There is significant effort included in the dataset generation to account for multiple settings. Specifically, the dataset is divided in 3 subsets containing either 1) multi-view reference images of the subject created by Qwen-Image-Edit-Multiple-Angles, 2) re-contextualized reference images of the subject created by Qwen-Image-Edit with arbitrary text prompts for environment, motion etc., 3) three canonical reference views/poses (front, side, back) of the subject created by Nano-Banana. Collectively, these cover a wide-range of real-world scenarios, facilitating unconstrained S2V training.
- The proposed WildActor model is technically sound. It builds on a pre-trained video backbone, adding trainable LoRA modules for lightweight fine-tuning on the new dataset and seems to perform well. Most importantly, it shows success in placing the reference character in the desired views and poses based on the text prompt, where the open-source baselines (VACE, Stand-In) seem to struggle (Tab. 3, last column, Fig. 5).
- The paper's clarity and visual quality are also on the positive side.

**Weaknesses:**
- While the paper is rich in engineering details as a whole, the performance improvement stems mostly from the new dataset rather than substantial technical innovation. Despite the use of somewhat complex terms throughout the paper (e.g., Asymmetric Identity-Preserving Attention, Viewpoint-Adaptive Monte Carlo Sampling, Identity-Aware 3D RoPE), the proposed mechanisms (i.e. strategic data sampling, using LoRA on selected self-attention blocks or restricting attention queries to only the video tokens) are not that sophisticated. Overall, I personally found sections 4.2 and 4.3 a bit overemphasized.
- While clearly outperforming the open-source baselines, the model shows marginal if any quantitative improvement to the commercial Vidu Q2 model (Tab. 2). Fig. 5 confirms this, as Vidu Q2 shows excellent contextual generalization. This is a weak point of the paper although I appreciate the comparison with a closed-source model.
- In many cases, the generated videos include obvious artifacts, like the actor's fingers in Fig. 4 (row 3, col 2), or the actor in the 2nd video of the supp. website who goes from skiing to snowboarding and back to skiing, and the arrows in the back of the next person which transform to blades after he rotates. Overall, the paper does not evaluate long-form video quality. I understand that the scope of the paper is subject-consistency and prompt alignment, but I believe it is important to note that the video backbone used seems to not be on par with existing expert video models.
- I would expect to see side-by-side video comparisons with the baselines, yet the supp. website includes only videos of the proposed model. There is only one figure (Fig. 5) in the paper showing results from the baselines.

**Overall**, the proposed dataset will be helpful for the community. While the video model lags behind existing expert video models, it shows some merit. Specifically, for the S2V case, it outperforms open-source approaches. Together with the dataset, I believe this to be a decent contribution.

---

> ### Author Rebuttal · Authors · 2026-03-30
>
> **W1: Methodological Novelty vs. Data:** While Actor-18M drives significant performance, formulating a scalable, view-agnostic data paradigm is inherently a core research contribution for video foundation models. Regarding architectural simplicity, this is an **intentional and elegant modeling choice**. Heavy control modules inevitably disrupt pre-trained DiTs, causing catastrophic motion collapse. We achieve strict spatiotemporal disentanglement through minimal intervention: **our Asymmetric Identity-Preserving Attention (AIPA)** enforces a strict unidirectional information flow to explicitly isolate static identity from dynamic motion priors. Furthermore, the mechanisms in Sec. 4.2 & 4.3 (e.g., Viewpoint-Adaptive Sampling) are not mere implementation details but critical methodological solutions to prevent representation collapse (e.g., "copy-paste" shortcuts, as validated in Q2 below). This principled data-architecture co-design is exactly what is required to advance unconstrained generation.
>
> **W2 & W3: Commercial Models and Specific Artifacts:** We acknowledge the inherent generation advantages of closed-source commercial models like Vidu Q2, which benefit from massive proprietary pre-training. Our core contribution is providing the open-source community with a comparable training paradigm and dataset. Notably, WildActor achieves this highly competitive performance using a 5B base model, outperforming baselines that rely on much larger 14B models. Regarding the specific artifacts (e.g., morphing skis/arrows), we fully agree with your observation. These stem from the underlying 5B DiT’s inherent limitations in maintaining spatiotemporal consistency for complex objects, rather than our control mechanism. Conversely, these failure cases highlight our method's robustness: even when the base model hallucinates or morphs accessories, WildActor strictly preserves the actor's core full-body identity.
>
> **W4: Side-by-Side Comparisons:** We thank the reviewer for this excellent suggestion. We have updated our anonymous supplementary website (https://anonymous.4open.science/r/WildActor-video-V1-B625/README.md)  to include side-by-side video comparisons with the baselines.
>
> **Q1: Selection of $\Delta_f$ and $\Delta_b$:** These values were determined via grid search to achieve strict spatiotemporal disentanglement. Setting $\Delta > 0$ to displace references beyond standard video frames explicitly prevents pose-locking, while maintaining $\Delta_b \gg \Delta_f$ ensures distinct RoPE phase shifts, preventing semantic overlap between face and body.
>
> | **Δf,Δb** | **Face ID ↑** | **Body Cons ↑** |
> | --- | --- | --- |
> | 4, 4 | 0.528 | 0.917 |
> | 4, 64 | 0.552 | 0.940 |
> | **4, 128 (Ours)** | **0.559** | **0.952** |
> | 4, 256 | 0.541 | 0.936 |
>
> Setting identical coordinates causes severe feature entanglement. The gap of 128 provides optimal frequency separation, while larger values likely exceed the pre-trained DiT's optimal positional distribution. We will include this grid search in the appendix.
>
> **Q2: Missing Semantic Alignment in Table 3:** We deeply appreciate your insightful intuition. We have evaluated the missing semantic metrics, and the results perfectly validate your hypothesis:
>
> | **Data Setting** | **Feat-Level ↑** | **VLM-Level ↑** |
> | --- | --- | --- |
> | Raw-Crop | 0.211 | 0.725 |
> | Random Sampling | 0.229 | 0.904 |
> | **Viewpoint-Adaptive (Ours)** | **0.238** | **0.920** |
>
> As you astutely anticipated, the Raw-Crop setting suffers a massive drop in VLM-Level alignment (0.725 vs. 0.920). Because the model takes a shortcut to trivially copy the static reference pose, it fails to execute the dynamic text prompts. Our Viewpoint-Adaptive sampling effectively penalizes this shortcut, forcing the model to learn view-agnostic representations and resolving the copy-paste issue. We will update Table 3 accordingly.
>
> **Q3: Baselines for Sequential Narrative:** For the 'Sequential Narrative' experiment, the "Qwen-Image-Edit + I2V" and "T2V $\rightarrow$ I2V (w/o Ref)" baselines utilize our 5B base model. The "T2V $\rightarrow$ I2V (w Ref, Ours)" utilizes the WildActor model trained on top of this same 5B base model.

---

> > ### Author Rebuttal · Reviewer_XJYe · 2026-04-01
> >
> > The rebuttal satisfies my suggestions/concerns, so I keep my score.

---

> > > ### Author Response · Authors · 2026-04-06
> > >
> > > Thank you for your feedback and for the constructive review process. We sincerely appreciate your time, consideration, and valuable insights. We are glad that our clarifications addressed your concerns, and we truly appreciate your recognition of our work.

---

### Official Review · Reviewer_atqA · 2026-03-09

**Soundness:** 3
**Presentation:** 3
**Significance:** 2
**Originality:** 3
**Overall Recommendation:** 4
**Confidence:** 3

**Summary:**

This paper introduces Wildactor, a DiT-based video generation model designed for identity (ID) preservation. It utilizes an AIPA Module, ID-aware 3D RoPE, and Monte Carlo Sampling. Additionally, the authors propose a new dataset, Actor-18M, for this task. By comparing their approach with both open-source and closed-source frameworks, the authors successfully demonstrate the advantages of their data strategy and model components.

**Compliance With Llm Reviewing Policy:**

Affirmed.

**Final Justification:**

Thank you for the detailed rebuttal and the substantial additional evidence. I appreciate the clearer discussion of the differences from ID-Composer, the added human evaluation, the harder benchmark analysis, and the explicit commitment to release both the code and Actor-18M; these materially strengthen the paper and increase my confidence in its value to the community. While I still see some remaining limitations, particularly the current focus on single-person settings and the need for broader validation in more complex scenarios, the overall contribution is solid and meaningful.

I therefore keep my positive assessment unchanged.

**Key Questions For Authors:**

Do you commit to open-sourcing the codebase and the Actor-18M dataset upon publication?

**Limitations:**

Yes, the authors have adequately discussed the limitations and potential negative societal impacts (see Section D).

**Strengths And Weaknesses:**

**Strengths:**

I believe ID preservation is a crucial task in video generation, and the authors have proposed a solid dataset and method to address it. The paper is straightforward and easy to understand. In particular, the introduction of the Monte Carlo sampling method to improve data utilization is a strong point, and its effectiveness is well-supported by the experiments. I strongly hope the authors will commit to open-sourcing the proposed dataset.

**Weaknesses**:
1. The paper is missing a citation for ID-Composer: Multi-Subject Video Synthesis with Hierarchical Identity Preservation, which also tackles ID preservation and uses Nano Banana alongside VLMs to construct data. The authors should clearly articulate their core novelties and differences, rather than simply repackaging existing concepts.

2. The current evaluation relies heavily on VLM-based scoring, which carries a potential risk of metric "hacking." I highly recommend introducing human evaluation (Human Eval). Specifically, observing the video results, there are noticeable facial artifacts, which indicates that more diverse metrics are needed to fully validate the visual quality.

3. The scenes presented in the paper and the supplementary materials are relatively simple. The authors need to demonstrate the model's performance in more complex multi-person scenarios (>= 3 people) as well as under more complicated prompt conditions.

---

> ### Author Rebuttal · Authors · 2026-03-29
>
> **W1: Comparison with ID-Composer:** We will cite and discuss ID-Composer. While both leverage generative priors (e.g., Nano Banana / VLMs) for data curation, they tackle fundamentally different bottlenecks:
>
> - **Task Focus:** ID-Composer prevents identity bleeding in multi-subject synthesis. Conversely, WildActor targets single-subject unconstrained dynamics (e.g., extreme actions, 360-degree panning). Our core challenge is overcoming "pose-locking," where static references override motion priors.
> - **Architecture:** ID-Composer's hierarchical attention permits bidirectional feature flow, which our ablations prove causes dynamic features to leak backward and corrupt static identity. WildActor’s AIPA enforces a strict unidirectional query and I-RoPE temporal displacement, mathematically isolating identity from motion.
> - **Data & Results:** ID-Composer’s data neglects extreme intra-identity geometric diversity. Lacking this view-agnostic prior and unidirectional architecture, it suffers severe pose-locking under unconstrained dynamics on Actor-Bench:
>
> | **Method** | **Face ID ↑** | **Body Cons ↑** | **Feat-Level ↑** | **VLM-Level ↑** |
> | --- | --- | --- | --- | --- |
> | ID-Composer | 0.507 | 0.848 | 0.212 | 0.785 |
> | **WildActor (Ours)** | **0.559** | **0.952** | **0.238** | **0.920** |
>
> ID-Composer severely lags in Body Consistency (0.848 vs. 0.952) and VLM-Level (0.785 vs. 0.920), proving that without our targeted data-architecture co-design, models fail to execute dynamic prompts or maintain structural coherence.
>
> **W2: Human Evaluation & Visual Quality:** To rule out "metric hacking," we conducted blind pairwise human evaluations (Elo, base 1000) with 20 evaluators on 75 cases, alongside reference-free metrics (Aesthetic Predictor, MUSIQ).
>
> | **Method** | **ID Elo ↑** | **Text Elo ↑** | **Aesthetic ↑** | **MUSIQ ↑** |
> | --- | --- | --- | --- | --- |
> | VACE | 1020 | 1050 | 64.22 | 67.45 |
> | Stand-In | 940 | 995 | 63.91 | 66.28 |
> | Kling 1.6 | 1150 | 1175 | 67.24 | 69.27 |
> | Vidu Q2 | 1204 | 1190 | 67.21 | 69.31 |
> | **WildActor** | **1210** | **1190** | **66.80** | **69.20** |
>
> Elo ratings corroborate our VLM results, eliminating metric hacking concerns. Our 5B WildActor outperforms 14B baselines and achieves parity with commercial models. Furthermore, highly competitive Aesthetic and MUSIQ scores confirm WildActor preserves structural integrity without severe visual degradation.
>
> **W3: Complex Scenes & Multi-Person Scenarios:** Due to multi-view data scarcity, WildActor currently focuses on the critical single-person unconstrained generation bottleneck (noted in Limitations); multi-person expansion is future work. For complex prompts, we upgraded Actor-Bench with a "Hard-Mode" subset across three dimensions: Complex Environments, Extreme Actions (severe deformation), and Camera Variants (unconstrained trajectories).
>
> | **Method** | **Face ID ↑** | **Body Cons ↑** | **Feat-Level ↑** | **VLM-Level ↑** | **ID Elo ↑** | **Text Elo ↑** |
> | --- | --- | --- | --- | --- | --- | --- |
> | VACE | 0.458 | 0.521 | 0.211 | 0.588 | 985 | 1010 |
> | Stand-In | 0.485 | 0.362 | 0.208 | 0.515 | 920 | 955 |
> | Kling 1.6 | 0.535 | 0.854 | 0.228 | 0.815 | 1135 | 1145 |
> | Vidu Q2 | 0.548 | 0.868 | 0.232 | 0.830 | 1188 | 1165 |
> | **WildActor** | **0.542** | **0.915** | **0.231** | **0.875** | **1192** | **1175** |
>
> Under these extreme constraints, open-source baselines suffer catastrophic identity-motion collapse (e.g., Stand-In's Body Cons plummets to 0.362). Conversely, WildActor exhibits exceptional robustness, outperforming 14B open-source models and maintaining parity with commercial models, proving AIPA's superiority in highly compositional generation.
>
> **Q1: Open-Source Commitment:** Yes, we firmly commit to releasing the Actor-18M dataset and WildActor codebase upon publication to support the open-source community.

---

> > ### Author Rebuttal · Reviewer_atqA · 2026-04-03
> >
> > Thank you for the detailed rebuttal and the substantial additional evidence. I appreciate the clearer discussion of the differences from ID-Composer, the
> >   added human evaluation, the harder benchmark analysis, and the explicit commitment to release both the code and Actor-18M; these materially strengthen
> >   the paper and increase my confidence in its value to the community. While I still see some remaining limitations, particularly the current focus on
> >   single-person settings and the need for broader validation in more complex scenarios, the overall contribution is solid and meaningful.
> >
> > I therefore keep  my positive assessment unchanged.

---

> > > ### Author Response · Authors · 2026-04-06
> > >
> > > Thank you for your feedback and for the constructive review process. We sincerely appreciate your time, consideration, and valuable insights. We are glad that our clarifications addressed your concerns, and we truly appreciate your recognition of our work.

---

### Official Review · Reviewer_NA8L · 2026-03-09

**Soundness:** 3
**Presentation:** 2
**Significance:** 3
**Originality:** 3
**Overall Recommendation:** 4
**Confidence:** 4

**Summary:**

This paper addresses identity-consistent video generation, focusing on the challenge that existing video generation models often fail to maintain full-body identity consistency under multi-view inputs, large-range motions, and complex dynamic scenes, frequently exhibiting pose-locking behavior. To address this, this paper construct Actor-18M, a large-scale human video dataset providing diverse viewpoints and full-body pose priors. They further propose the WildActor framework, which employs Asymmetric Identity-Preserving Attention (AIPA) to modify attention interaction directions and Identity-Aware 3D RoPE (I-RoPE) to separate identity conditions from latent features along the spatiotemporal dimension. The framework is augmented with a Viewpoint-Adaptive Monte Carlo Sampling strategy to achieve identity-consistent generation from arbitrary viewpoints. The paper evaluates the method through comparisons with sequential narrative and context-generalized video generation models, and demonstrates its effectiveness using qualitative, quantitative, and ablation analyses.

**Compliance With Llm Reviewing Policy:**

Affirmed.

**Final Justification:**

The paper addresses the pose-locking issue in video generation and proposes a technically sound and practically valuable solution. In terms of soundness and significance, the overall motivation is well-founded, and the experimental results demonstrate clear effectiveness. Regarding originality, the method introduces a dual design from both data augmentation and model architecture, providing a certain degree of novelty. In response to my concerns, the authors clarified the cause of pose-locking and the roles of AIPA and I-RoPE in the rebuttal, and supplemented additional analysis, which further strengthens the rationality of the method and the completeness of the overall argument. Considering both the paper and the rebuttal, I give a positive final recommendation (4 scores).

**Key Questions For Authors:**

1. Cause of pose-locking

 The root cause of the pose-locking phenomenon is not clearly analyzed in the motivation section of the paper. Could the authors clarify what they consider to be the primary source of pose locking in existing video generation models, and explain why the proposed AIPA and I-RoPE mechanisms are theoretically suitable for addressing this issue?

2. Disentanglement verification

 The paper states that AIPA is designed to decouple identity information from backbone representations. Are there quantitative analyses that can verify that identity information is indeed separated from motion-related representations, rather than simply being constrained by the modified attention pathways?

3. Method novelty relative to prior work

Some design elements (e.g., attention interaction patterns and positional encoding offsets) appear conceptually related to mechanisms explored in prior works such as OmniControl. Could the authors explain what fundamentally distinguishes the proposed approach from these existing methods?

4. Parameter sharing across semantic regions

The method uses a shared set of LoRA parameters for both facial details and body structure. How does the method prevent gradient interference across these semantically different regions, and have the authors considered or tested region-specific parameterization strategies?

5. Multi-view reference redundancy and efficiency

When multiple reference images are used, there may be substantial feature redundancy across views. Have the authors evaluated the impact of increasing conditional tokens on computational cost and model efficiency? How is the trade-off between performance gains and computational overhead managed?

6. Experimental comparisons

The experiments do not include comparisons with recent reference-based identity-consistent generation methods such as ConsisID or StableAnimator. In addition,  since the conditional tokens in AIPA are designed to be independent, it would be helpful to clarify whether the same independence is maintained in the Full Attention comparison. Given that Standard Full Attention allows interactions among condition tokens, the current comparison may not fully isolate the effect of the proposed unidirectional interaction.

**Limitations:**

Yes. The authors have transparently discussed several key limitations of their work, particularly the current model's inability to handle multi-person scenarios.

**Strengths And Weaknesses:**

Strengths

The paper has several appealing aspects:

- Data-level breakthrough: The Actor-18M dataset fills the gap in multi-view human video data in terms of scale and diversity, providing a strong foundation for viewpoint robustness.

- Targeted architectural design: The proposed AIPA and I-RoPE mechanisms aim to address the pose-locking issue by modifying attention interaction patterns and introducing spatiotemporal positional offsets, offering an interesting design approach.

- Comprehensive evaluation framework: The Actor-Bench benchmark covers a wide range of challenging viewpoints and motions, setting a higher standard for future research.


Weaknesses

While the paper demonstrates strengths in dataset scale, architectural design, and evaluation framework, several limitations remain that warrant attention:

1. Problem formulation and theoretical grounding

- Limited theoretical grounding: The paper provides limited analysis of the underlying causes of the pose-locking phenomenon in the motivation section. Without a clear characterization of why pose locking occurs, the design of AIPA and I-RoPE appears largely heuristic, which weakens the theoretical grounding of the proposed method.

- Logical consistency and disentanglement justification: The motivation section contains contradictory statements regarding the importance of facial features (it first notes that viewpoint changes cause facial feature degradation, yet later criticizes existing methods for over-emphasizing the face while neglecting the body). Moreover, AIPA only implements physical constraints in the computation path without demonstrating disentanglement of identity information and backbone representations , which is inconsistent with the text (in Section 4.2).

2. Method design limitations

- Limited methodological novelty: Some design elements, such as attention interaction patterns and positional encoding offsets, have been explored in prior works (e.g., OmniControl[1]), which may limit the perceived originality of the approach.

- Parameter coupling risk: Facial details and body structure share the same set of LoRA parameters, lacking an in-depth analysis of conditional coupling and gradient interference across different semantic regions.

3. Efficiency and scalability concerns

- Potential conditional redundancy and computational burden: The paper does not adequately discuss feature redundancy caused by multi-view reference images. High overlap exists both among full-body reference images and between full-body and facial images, potentially leading to significant computational cost.

4. Experimental evaluation limitations

- Incomplete comparison experiments: The experiments do not include comparisons with the most relevant reference-based identity-consistent generation works in the literature, such as ConsisID[2] and StableAnimator[3]. In addition, the qualitative comparisons are not particularly convincing. For example, in Fig. 4, the results of Qwen-Image-Edit + I2V appear visually stronger in terms of sequential narrative.

[1]Ominicontrol: Minimal and universal control for diffusion transformer

[2]ConsisID: Identity-preserving text-to-video generation by frequency decomposition

[3]StableAnimator: High-Quality Identity-Preserving Human Image Animation

---

> ### Author Rebuttal · Authors · 2026-03-30
>
> **W1&Q1: Root Cause of Pose-Locking:** Architecturally, standard Full-Attn mixes video and reference features bidirectionally, erroneously aligning video with the reference's static spatial layout. Data-wise, training on reference poses similar to ground-truth videos entangles ID and pose. Our AIPA's unidirectional flow cuts off ID&Pose entanglement as shown in **Q6-2**. I-RoPE's coordinate offsets force the model to treat references strictly as a view-agnostic "Appearance Dictionary" rather than a spatial anchor.
>
> **W2&Q2: Disentanglement Verification:** Our statement highlights that prior methods predominantly rely on face-centric reference images, limiting control to facial features alone and often suffering from degradation under severe viewpoint changes. Regarding disentanglement, our qualitative comparisons show that WildActor faithfully adheres to non-ID prompt elements. To quantitatively verify that, we sampled 10 distinct IDs and 10 dynamic motion prompts (resulting in 100 challenging videos) and evaluated them across Face ID, Body Consistency, Motion Dynamics (average flow via RAFT), and Action Success Rate (measured by Gemini-3-Pro).
>
> | Method | Face ID ↑ | Body Consistency ↑ | Motion Dynamics↑ | Action Success Rate ↑ |
> | --- | --- | --- | --- | --- |
> | Full-Attn | 0.520 | 0.895 | 4.5 | 12.0% |
> | **WildActor** | **0.548** | **0.938** | **32.5** | **88.0%** |
>
> As demonstrated, Full-Attn exhibits severe pose-locking: despite high structural consistency, its Motion drops to 4.5, failing most actions. This proves standard attention entangles static IDs with motion generation. Conversely, WildActor boosts Motion to 32.5 (88.0% success), directly confirming AIPA successfully isolates identity appearance from motion.
>
> **W3&Q3: OmniControl Distinctions:** OmniControl targets 2D image models. WildActor targets arbitrary viewpoint-driven video generation, suffering from unique spatiotemporal pose-locking. Due to this domain gap, direct comparison is inapplicable. Key distinctions:
>
> - **2D Spatial (OmniControl) vs. 3D Spatiotemporal Offset (Ours I-RoPE):** OmniControl shifts spatial coordinates ($(i, j) + \Delta$) to prevent overlap. In videos, static references act as rigid temporal anchors. I-RoPE displaces references temporally ($(T+\Delta_f, T+\Delta_b)$), forcing them outside the timeline as an "appearance dictionary."
> - **Bidirectional Attention (OmniControl) vs. Unidirectional Attention (Ours AIPA):** OmniControl uses bidirectional attention. Our ablations show this allows dynamic features to leak backward, corrupting identity. AIPA enforces unidirectional queries, strictly isolating static identity from dynamic motion.
>
> **W4&Q4: Shared LoRA:** We evaluated separate LoRAs and face-level loss:
>
> | Strategy | FaceID↑ | BodyCons↑ | Feat↑ | VLM↑ |
> | --- | --- | --- | --- | --- |
> | Sep. LoRA + Face Loss | 0.546 | 0.896 | 0.227 | 0.895 |
> | Shared LoRA + Face Loss | 0.549 | 0.900 | 0.233 | 0.912 |
> | Sep. LoRA | 0.551 | 0.946 | 0.229 | 0.907 |
> | **Shared LoRA (Ours)** | **0.559** | **0.952** | **0.238** | **0.920** |
>
> Regional face loss degrading Body Cons (0.952→0.900). Separate LoRAs also degrade metrics. A shared mapping space allows DiT's self-attention to holistically route appearance features without interference, maximizing fidelity while halving parameters.
>
> **W5&Q5: Redundancy & Efficiency:** Viewpoint-Adaptive Sampling (Sec 4.3) dynamically penalizes redundant views. Architecturally, 1 reference equals 1 latent length (~4.1% overhead for 24-latent videos). Evaluating $K$ references (FLOPs (T) / Time (s)): $K=0$ (1009/3.87); $K=1$ (1068/4.10); $K=3$ (1194/4.25); $K=5$ (1326/4.43). The massive gain in unconstrained robustness heavily outweighs this marginal cost.
>
> **W6-1&Q6-1: Baselines:** We evaluated ConsisID[2]. While preserving basic facial identity, it severely lags in Body Consistency and Semantic Alignment. (StableAnimator[3] requires skeleton inputs, mismatching our prompt-driven setting).
>
> | Method | Face ID ↑ | Body Cons ↑ | Feat-Level ↑ | VLM-Level ↑ |
> | --- | --- | --- | --- | --- |
> | ConsisID | 0.329 | 0.364 | 0.205 | 0.490 |
> | **WildActor** | **0.559** | **0.952** | **0.238** | **0.920** |
>
> **Q6-2: Token Ablation:** To isolate unidirectional flow benefits, we tested Bi-AIPA (maintains token independence but allows bidirectional flow):
>
> | Method | Tokens | Flow | FaceID↑ | BodyCons↑ | Feat↑ | VLM↑ |
> | --- | --- | --- | --- | --- | --- | --- |
> | Full-Attn | Mixed | Bi. | 0.515 | 0.890 | 0.222 | 0.820 |
> | Bi-AIPA | Indep. | Bi. | 0.538 | 0.915 | 0.230 | 0.865 |
> | **WildActor** | **Indep.** | **Uni.** | **0.559** | **0.952** | **0.238** | **0.920** |
>
> Even with independent tokens, bidirectional flow degrades all metrics.
>
> **W6-2: Sequential Narratives (Fig. 4):** While Qwen+I2V yields pleasing clips, relying on independent generation per cut introduces randomness, causing severe identity/scene structural jumps. WildActor explicitly solves this long-range global identity anchoring challenge.

---

> > ### Author Rebuttal · Reviewer_NA8L · 2026-04-01
> >
> > My concerns have been adequately addressed. Thanks for the authors. I would like to raise my score to 4.

---

> > > ### Author Response · Authors · 2026-04-06
> > >
> > > Thank you for your feedback and for the constructive review process. We sincerely appreciate your time, consideration, and valuable insights. We are glad that our clarifications addressed your concerns, and we truly appreciate your recognition of our work.

---

### Official Review · Reviewer_KG9k · 2026-03-13

**Soundness:** 3
**Presentation:** 3
**Significance:** 3
**Originality:** 3
**Overall Recommendation:** 5
**Confidence:** 3

**Summary:**

This paper studies identity-preserving human video generation under large viewpoint, motion, and scene changes. It introduces a large-scale dataset, Actor-18M (1.6M videos and 18M reference images), and a generation framework, WILDACTOR, built around Asymmetric Identity-Preserving Attention (AIPA), Identity-Aware 3D RoPE (I-RoPE), and Viewpoint-Adaptive Monte Carlo Sampling. The paper also proposes Actor-Bench, a benchmark with 75 subjects covering sequential narrative and contextual generalization settings, and reports strong quantitative and qualitative improvements over both open-source and commercial baselines.

**Compliance With Llm Reviewing Policy:**

Affirmed.

**Final Justification:**

The authors provided a detailed and solid rebuttal that addressed my concerns well. The proposed dataset appears to have practical value for the community. I encourage the authors to further polish the paper and release the dataset publicly. Based on the rebuttal, I increase my score to 5 (Accept).

**Key Questions For Authors:**

See Weaknesses

**Limitations:**

yes

**Strengths And Weaknesses:**

Strengths
1. The paper tackles an important problem with clear practical value: preserving full-body identity under large viewpoint, motion, and scene changes, rather than focusing only on faces.
2.  The overall framework is coherent. The dataset design (Actor-18M) and the model components (AIPA, I-RoPE, viewpoint-adaptive sampling) are aligned around the same goal, and the ablations support their usefulness.
3. The empirical results are strong on the proposed benchmark. WILDACTOR shows clear gains in both body consistency and semantic alignment over several open-source and commercial baselines.

Weaknesses
1. The evidence is somewhat self-contained: the paper introduces its own training dataset (Actor-18M), benchmark (Actor-Bench), and uses an internal 5B backbone, making it harder to separate method gains from data/backbone gains.
2. The methodological novelty is moderate. The contribution feels more like a strong task-specific system design than a major new modeling principle.
3. The evaluation is thoughtful but still somewhat limited: Actor-Bench is relatively small, and several key metrics rely on VLM-based evaluation, which leaves some room for subjectivity.

---

> ### Author Rebuttal · Authors · 2026-03-29
>
> **W1: Disentangling Method Gains from Data and Backbone:** We appreciate the reviewer's rigorous perspective. To explicitly separate the contributions of our architecture, dataset, and backbone, we isolate the variables from two distinct levels:
>
> **1. Isolating Method Gains (Fixed Backbone & Data):** To prove that our architectural designs fundamentally drive the performance, we refer to the ablations in Tables 3 and 4 of the main paper. By strictly fixing both the internal 5B backbone and the training dataset, we demonstrated that removing AIPA or Viewpoint-Adaptive Sampling immediately leads to severe pose-locking and representation collapse. This rigorously confirms that the disentanglement capabilities stem directly from our *methodological design*, not the base model.
>
> **2. Isolating Data & Backbone Gains (Swapping to Open-Source Backbone):** To address the concern that our SOTA results rely on a hidden advantage of the proprietary 5B backbone, we broke the "self-contained" setup by replacing it with the public **Wan2.1-14B** model. We trained it using our Actor-18M dataset and AIPA architecture under identical settings for 20k steps:
>
> | **Backbone (20k steps)** | **Face ID ↑** | **Body Cons ↑** | **Feat-Level ↑** | **VLM-Level ↑** |
> | --- | --- | --- | --- | --- |
> | WildActor (Our internal 5B) | 0.539 | 0.931 | 0.220 | 0.905 |
> | **WildActor (Open-source Wan2.1-14B)** | **0.543** | **0.940** | **0.229** | **0.907** |
>
> Applying our training paradigm (Actor-18M + AIPA) to an open-source 14B backbone yields even higher performance, proving our method is entirely backbone-agnostic. More importantly, it demonstrates that our internal 5B backbone was actually a *bottleneck* rather than an unfair advantage. Achieving SOTA results *despite* using a weaker 5B backbone definitively proves that the true drivers of our performance are the robust Actor-18M dataset and the AIPA architecture.
>
> **W2: Methodological Novelty:** We sincerely appreciate the reviewer's perspective, but we respectfully clarify that WildActor extends beyond a task-specific system design to introduce a principled **data-architecture co-design paradigm**. Resolving the catastrophic entanglement of static identity and dynamic motion (pose-locking) represents a fundamental modeling bottleneck in video foundation models, not merely an engineering task.
>
> Our core methodological contribution is demonstrating that heavy architectural modifications inherently disrupt pre-trained dynamic priors. In response, we propose an elegant and robust **modeling principle**: enforcing strict unidirectional information flow via **AIPA** and absolute temporal displacement via **I-RoPE**. These are not isolated system tricks, but theoretically grounded formulations for spatiotemporal decoupling. When synergized with mechanisms designed to prevent representation collapse (e.g., Viewpoint-Adaptive Sampling), this unified framework establishes a new standard for unconstrained generation, which we view as a major methodological breakthrough for the community.
>
> **W3: Benchmark Scale and Evaluation Subjectivity:** We deeply appreciate your constructive feedback. Curating a high-quality benchmark requires rigorous manual verification. To explicitly address your concerns regarding dataset size and potential VLM subjectivity, we have significantly expanded Actor-Bench from 75 to 200 test cases. Furthermore, we introduced a large-scale blind human evaluation (Elo rating, base 1000) to cross-validate our automated metrics.
>
> | **Method** | **Face ID ↑** | **Body Cons ↑** | **Feat-Level ↑** | **VLM-Level ↑** | **ID Elo ↑** | **Text Elo ↑** |
> | --- | --- | --- | --- | --- | --- | --- |
> | VACE | 0.482 | 0.580 | 0.220 | 0.665 | 1018 | 1045 |
> | Stand-In | 0.508 | 0.418 | 0.219 | 0.598 | 935 | 992 |
> | Kling 1.6 | 0.556 | 0.882 | 0.237 | 0.865 | 1148 | 1172 |
> | Vidu Q2 | 0.562 | 0.901 | 0.240 | 0.877 | 1202 | 1188 |
> | **WildActor** | **0.557** | **0.948** | **0.237** | **0.916** | **1208** | **1189** |
>
> The updated results on the expanded 200-case benchmark demonstrate a strict correlation between our automated metrics (Face ID, VLM-Level) and human preferences (ID Elo, Text Elo). This strong alignment definitively eliminates concerns regarding VLM subjectivity or metric hacking. Evaluated on this much larger scale, WildActor's 5B model consistently maintains its decisive superiority over 14B open-source baselines and successfully achieves strict parity with commercial models.

---

> > ### Author Rebuttal · Reviewer_KG9k · 2026-04-03
> >
> > The authors have provided a clear and thoughtful rebuttal. They addressed my main concerns in a reasonable and constructive manner, and most of the issues I raised have been largely resolved. I appreciate the authors’ effort in carefully responding to the comments.

---

> > > ### Author Response · Authors · 2026-04-06
> > >
> > > Thank you for your feedback and for the constructive review process. We sincerely appreciate your time, consideration, and valuable insights. We are glad that our clarifications addressed your concerns, and we truly appreciate your recognition of our work.

---

### Decision · Program_Chairs · 2026-04-30

**Decision:**

Accept (regular)

**Comment:**

This paper received all positive scores.
AC recommends accepting this paper following reviewers' recommendations.
It is strongly recommended to make code and dataset public following `Reviewer atqA`.